# COMPOTE: GENERATING A DATASET OF REAL-WORLD BINARY LEVEL VULNERABILITIES

## ABSTRACT

Once a proprietary program written in a compiled language like `C` is successfully compiled, it is typically distributed as a binary executable. Consequently, security analysis of the program, including vulnerability detection, relies solely on the binary. Binary-level detection methods have been developed over the years, with machine learning (ML)-based methods becoming increasingly popular. However, the scarcity of high-quality, publicly available datasets limits the development of ML-based binary vulnerability detectors, as existing binary-level vulnerability datasets are often synthetic and fail to reflect real-world vulnerabilities. At the same time, existing real-world source-code vulnerability datasets cannot be directly compiled, as they typically consist of standalone function snippets rather than compilable programs. To address this limitation, we present **Compote**, a **COMP**ilation AI-**O**rchestrated **T**ransformation **E**ngine that automatically wraps standalone `C` functions with the minimal scaffolding, such as headers, mocks, and `main()`, needed for successful compilation of `C` functions without altering the original code. Applying Compote to real-world functions from ten public datasets of vulnerable code yields a dataset comprising 18K compilable `C` functions along with their compiled binary versions. Our dataset represents a novel, large-scale, realistic, labeled benchmark spanning both source and binary domains. To evaluate our dataset, we fine-tune state-of-the-art vulnerability detection models. We show that models trained and tested exclusively on existing (synthetic) datasets achieve up to 98.97% $F_1$ but drop to 29.28% when tested on the real-world vulnerabilities in our dataset. This demonstrates the inability of models trained on synthetic datasets to generalize effectively to real-world binary vulnerabilities, resulting in a significant drop in detection performance. We release Compote and our datasets to the research community to support further research on building and evaluating effective and practical binary vulnerability detection models.

## 1 INTRODUCTION

One of the main steps in software development with compiled languages is translating high-level code into machine-executable instructions - a process known as compilation (Wu & Tang, 2023). However, not all code segments are directly compilable without modification. Compiled languages like `C` require complete, self-contained programs with all supporting components (e.g., header files, mock functions, and include directives) correctly defined and available (Kabir et al., 2024). Nevertheless, these prerequisites create barriers when immediate compilation and execution of code snippets is necessary - such as during testing, analysis, or debugging. This complicates testing standalone source files and creates friction when working with incomplete contexts.

To address this challenge, we introduce *Compote*, an AI-based tool designed to automatically transform standalone C functions into fully compilable programs. Compote receives an isolated C code snippet and generates all the necessary wrapper code to satisfy compilation requirements - including a main() function, required headers, and mock definitions, producing a complete, self-contained C program that can be successfully compiled and executed. To the best of our knowledge, Compote is the first tool that automates this process using AI-driven workflow techniques while ensuring that the underlying function remains unchanged. Specifically, we implemented Compote as an AI-orchestrated workflow. This automation capability reduces manual effort and enables developers, researchers, and security analysts to compile and analyze code more efficiently.

While Compote is broadly applicable across many tasks such as function testing, automated binary debugging, and reverse engineering, one area where this automation can prove particularly useful is binary-level vulnerability detection. As machine learning (ML) advances, it has become a key method for automatically detecting vulnerabilities in code. However, ML models rely heavily on high-quality training data. In the context of binary-level vulnerability detection, the availability of suitable training datasets is severely limited: currently, only three main publicly available datasets exist Juliet (NSA Center for Assured Software, 2017), NDSS18 (Le et al., 2019) and BinPool (Arasteh et al., 2025). Juilet and NDSS18 datasets are semi-synthetic and lack real-world complexity, while the third Arasteh et al. (2025) is derived from real-world projects. See detailed explanation in Section 2.

The source code level vulnerability detectors benefit from a plethora of high-quality datasets derived from real-world projects Wang et al. (2021); Bhandari et al. (2021); Chakraborty et al. (2021); Zhou et al. (2019); Chen et al. (2023); Ding et al. (2024); Fan et al. (2020); Ni et al. (2024); Wang et al. (2024); He & Vechev (2023). While these datasets provide a realistic foundation for vulnerability detection research, they cannot be directly compiled, since they contain partial code snippets rather than complete programs that meet compilation requirements. This limitation prevents the use of these datasets to train binary-level vulnerability detectors. Compiling these datasets would create binary-level datasets, alleviating the scarcity of training sets at the binary level and potentially improving binary-level vulnerability detectors.

To bridge this gap between high-quality source-level vulnerability datasets and binary-level vulnerability detection, we applied Compote to process existing source-level datasets, thereby generating two new datasets: **CompRealVul_C** which consists of compilable C functions derived from source code snippets using Compote, labeled as vulnerable or non-vulnerable. **CompRealVul_Bin** contains the compiled binary versions of these functions, maintaining the vulnerability labels. Unlike semi-synthetic datasets (Juilet and NDSS18), CompRealVul_C and CompRealVul_Bin datasets are derived from real-world source code, ensuring they capture the complexity of actual software vulnerabilities.

To evaluate our approach, we fine-tuned and trained state-of-the-art binary-level vulnerability detection models using both the CompRealVul_Bin dataset and the widely used Juliet Test Suite (from the public GitHub repository Richardson (2024)). Our results show that models trained solely on synthetic datasets like Juliet struggle to generalize to real-world vulnerabilities, achieving only modest performance on realistic examples. In contrast, training on the Juliet Test Suite enriched with samples from CompRealVul_Bin led to consistent improvements in detection accuracy and generalization. This combined setup offered a more diverse and representative training environment, helping models with different architectures better distinguish between vulnerable and non-vulnerable code. Even when improvements were modest, they demonstrate the value of using the CompRealVul_Bin dataset as a more realistic benchmark for vulnerability detection.

We make the following contributions to the field: (1) We release two new datasets: *CompRealVul_C* and *CompRealVul_Bin* The former consists of real-world compilable C functions wrapped by Compote, while the latter contains their compiled binary versions. Both datasets include vulnerability labels, whilst *CompRealVul_C* includes also CWE indicators, supporting vulnerability research and detection. (2) We introduce *Compote*, an AI-based tool that automatically transforms standalone C source code snippets into compilable and executable binaries by generating the necessary wrapper code and build context (3) We demonstrate how Compote can help address the real-world problem of binary-level vulnerability detection by generating high-quality binary datasets from existing source-code level datasets.

## 2 BACKGROUND AND RELATED WORK

The automatic transformation of code snippets into compilable, self-contained units has long been an area of interest in both academic research and industry applications. In the context of the C programming language, this often involves wrapping isolated functions by inserting the necessary scaffolding: header files, type definitions, macros, and declarations that make the code compilable.

One approach to enabling such transformations is the use of automated tools designed to systematically modify C code according to predefined patterns or structural rules. For example, the creators of *Coccinelle* Padioleau et al. (2008) have devised the Semantic Patch Language (SmPL) to specify transformation rules in a syntax similar to C, enabling tasks like modifying function signatures, replacing deprecated patterns, and inserting missing structural elements. Le et al. (2019) proposed

an automated tool for detecting and fixing errors in incomplete C/C++ code snippets, allowing their compilation into binary functions. This method was used to construct a dataset of 32,281 binary functions across multiple platforms and architectures, derived from originally non-compilable code samples sourced from the NDSS18-SOURCE DATASET (Li et al., 2018). Another set of tools focuses on dependency resolution in C code. Clang-Include-Fixer (Team, 2016) suggests missing #include directives by analyzing unresolved symbols using a dedicated indexing and Abstract Syntax Tree; while Include What You Use (IWYU) (Project, 2011) helps ensure that only the necessary headers are explicitly included, removing any that are redundant.

Additionally, research has focused on inferring missing types and imports using program analysis or statistical methods. Subramanian et al. (2014) infers fully qualified names through constraint solving. Saifullah et al. (2019) improves ranking with statistical models based on context and naming patterns, though it remains probabilistic. SNR (Dong et al., 2022) combines constraint extraction with a library knowledge base and resolves imports using the Soufflé Datalog solver (Jordan et al., 2016), achieving 91.0% accuracy while compiling 73.8% of incomplete snippets.

A recent approach, ZS4C (Kabir et al., 2024), leverages LLMs to iteratively transform incomplete code snippets into compilable units through interactions with a compiler, achieving a 95.1% compilation success rate on Java and Python datasets. While effective, this method can unintentionally modify parts of the original code, potentially altering its intended behavior. In contrast, Compote explicitly preserves the original C function throughout the wrapping and compilation process, ensuring that the code semantics and any associated labels–such as vulnerability annotations–remain intact.

While prior work has addressed atomic challenges in making C code snippets compilable–such as inserting missing includes, resolving types and symbols, or fixing syntax and semantic errors–these capabilities are distributed across specialized tools. For instance, Clang-Include-Fixer (Team, 2016) and IWYU (Project, 2011) focus on header resolution, Coccinelle (Padioleau et al., 2008) supports structural code transformations, and Joern-based (Yamaguchi et al., 2014) tools repair incomplete syntax and semantics. However, no single tool offers a unified environment that systematically combines all these steps into a cohesive wrapping process. Compote addresses this gap by providing an all-in-one solution that automates the complete transformation pipeline, ensuring that C functions are preserved and transformed into compilable units without altering their core logic.

**Challenges in Binary-Level Vulnerability Detection.** Detecting vulnerabilities in binary code is a complex task in cybersecurity. Unlike source code analysis, which benefits from rich syntactic and semantic context, binary analysis operates on compiled executables where key information—such as variable names, data types, and control structures—is often lost or obfuscated (Zeng, 2012; Adhikari & Kulkarni, 2025; McCully et al., 2024). This lack of high-level context significantly increases the difficulty of identifying security flaws. Despite the fact that analyzing binary-level instructions is much harder compared to source code analysis, binary analysis remains critical for practical reasons, particularly when source-code is unavailable.

Although binary-level analysis provides substantial value, the progress of research in the area of vulnerability detection is often hindered by the lack of large-scale, compilable datasets that reflect diverse code patterns and structures. In recent years, ML has emerged as a promising approach for binary vulnerability detection with static analysis by learning patterns indicative of insecure behavior directly from binary code or its intermediate representations, such as assembly (Grieco et al., 2016; Shin et al., 2015). Recent papers have applied a range of ML approaches (Xu et al., 2017; Zhou et al., 2019); and transformer-based LLMs to this task (Brust et al., 2023).

The success of ML models in binary vulnerability detection is inherently tied to the quality of their training data. However, in the domain of binary vulnerability detection, a critical bottleneck remains: the lack of large-scale, realistic, and well-labeled datasets. The primary existing datasets, such as the Juliet Test Suite (NSA Center for Assured Software, 2017) and the NDSS18-compiled dataset (Le et al., 2019) are synthetic or semi-synthetic. Vulnerabilities in these datasets are generated according to specific templates (e.g., based on Common Weakness Enumerations - CWEs) or derived from simplified scenarios, limiting their diversity and realism. As a result, models trained exclusively on these datasets often show high benchmark performance but fail to generalize to real-world codebases (Chakraborty et al., 2021).

The Juliet Test Suite (NSA Center for Assured Software, 2017) includes 64,099 C/C++ test cases covering 118 CWEs and is designed for compilation. Test cases focus on demonstrating specific types

of flaws, though they may unintentionally include other unrelated issues. Alongside the flawed code, it typically includes similar, non-flawed code constructs for comparison. However, as a synthetic dataset, it lacks the variability and complexity of real-world code. Moreover, although each test case is meant to be unique, duplicates can arise during pre-processing, extraction, or compilation (Brust et al., 2023). These limitations restrict the depth of analysis and the practical applications the dataset can support, highlighting the need for a dataset that better captures real-world complexity and structural diversity. The NDSS18-compiled dataset (Le et al., 2019) comprises 32,281 functions compiled into 17,977 Windows binaries and 14,304 Linux binaries. However, to the best of our knowledge, it is only available in an embedded representation of the compiled samples, making it unsuitable for evaluations like ours that require access to the original binaries. A recent study Arasteh et al. (2025) compiled the Debian Project (2025) at the binary level, producing both pre-patch (vulnerable) and post-patch versions. The dataset also includes metadata identifying the specific file and function where each vulnerability occurs, and comprises 910 source-level functions and 7,280 corresponding binary functions. However, because the entire project was compiled as a single unit, only the full project binary is available after compilation.

In contrast, using our Compote, we generated the CompRealVul dataset, allowing each function to be compiled in isolation. To support this transformation, Compote, automatically wraps standalone C functions with the necessary elements for successful compilation, enabling rich, real-world source-code level datasets to be transformed into compilable binaries. Leveraging this tool, we introduce *CompRealVul_Bin*, a new dataset of real-world binary functions labeled for vulnerability detection. This approach bridges the compilation gap, supports realistic dataset generation, and lays the groundwork for more robust and accurate binary-level vulnerability detection models.

## 3  OUR METHOD

To facilitate the automated generation of compiled binaries from standalone C functions, Compote incrementally refines and validates code until it successfully compiles or a maximum iteration limit is reached. We implemented *Compote* as a workflow of transitions between state machines, some of which rely on an LLM for completion.

In our implementation, we used the GPT-4o-mini OpenAI (2024) API as the underlying LLM. This design enables breaking down the process into several modular components, making it straightforward to add, remove, or modify the system's logic. To transform standalone functions into compiled binary-ready code units, Compote's operation is broken down into four main components: (i) *Code Wrapping*; (ii) *Compilation and Validation*; (iii) *Error-Based Revision*; and (iv) *Function Calling*; as illustrated in Figure 1. The following subsections describe the main components of this process.

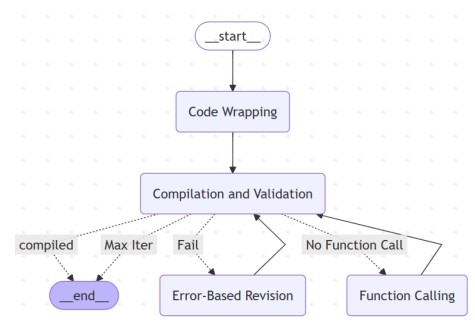

Figure 1: Compote workflow as a state machine.

**Code Wrapping.** The workflow begins by fetching a single C function, referred to as a "target function". Once fetched, a dedicated LLM model generates additional code, which, together with the target function, forms a minimal C program that meets compilation requirements. This includes adding necessary headers, declarations, stubs, and a main function with a call to the target function. The LLM is instructed to leave the function's intact, while ensuring compilability, as illustrated in Figure 2 that in Appendix Section B.1.

**Compilation and Validation.** Once the minimally compilable program code is generated, we need to ensure that the target function remains unchanged by the LLM. Although the LLM is instructed to preserve the content of the original function and only generate code around it (such as wrappers, headers, and the main function), in practice, the LLM may still modify the body of the function, either unintentionally or as part of an optimization attempt. Since preserving the functional integrity of the snippet is critical to maintaining its ground truth label (e.g., vulnerable or non-vulnerable), such modifications are not acceptable for our purposes. To prevent modifications to the target function, we overwrite its body in the minimally compilable program (returned by the *code wrapping* component)

with the original version of the target function. This ensures that the function being compiled is identical to the one extracted and labeled, thereby preventing the LLM from altering its semantics. This mechanism preserves the vulnerability characteristics of the minimally compilable program exactly as they appear in the target function, enabling reliable and reproducible binary-level results. At this stage, the *Compilation and Validation* component attempts to compile the resulting minimally compilable program. If compilation errors occur–ranging from syntax issues to missing symbols or type mismatches–the minimally compilable program, along with the compilation errors, is sent to the *error-based revision* component.

If no compiler errors are returned, the process results in a compiled binary. Before announcing a successful result, we ensure that the target function was called by the `main()` function of the minimally compilable program. If the target function was called, the process concludes. If not, the process continues by passing the minimally compilable program to the *Function Calling* component.

**Error-Based Revision.** If compilation fails, the process enters the *Error-Based Revision* phase. Here, the LLM revises the code , using two sources of context: (i) the last generated version of the minimally compilable program and (ii) the compiler error messages from the last compilation attempt. As illustrated in Figure 3 in Appendix Section B.1, at this stage the LLM is instructed to produce a targeted correction rather than regenerating the code from scratch.

**Function Calling.** A critical requirement of the process is that the code compiles and that the target function is executed in the resulting program. The *Function Calling* component addresses scenarios where the minimally compilable program compiles successfully, yet the validation step reveals that the target function is never invoked from the `main()` function. As illustrated in Figure 4 in Appendix Section B.1, the dedicated LLM of this phase is provided with the current minimally compilable program and is instructed to: (i) locate the existing `main()` function; (ii) insert a call to the original target function within `main()`; (iii) analyze the target function's signature to determine, declare, and initialize appropriate arguments for the call directly within `main()`. This step often requires the LLM to synthesize reasonable placeholder values or structures that satisfy the type requirements for successful compilation; (iv) strictly avoid modifying any other parts of the code, including the target function's body, existing statements in `main()`, or global definitions/includes. After the LLM of this component generates the modified code with the added function call, the process loops back to the *Compilation and Validation* phase.

**Putting it all together.** **Compote** takes each target function and cycles it through the steps illustrated in Figure 1. The process continues for every function until a successful result is produced or a predefined maximum number of iterations is reached (see Figure 5 in Appendix Section B.1).

## 4 CompRealVul Datasets

Existing datasets of vulnerable code can be broadly categorized into *compilable source code datasets* and *non-compilable source code datasets*. Compilable datasets, such as Juliet (NSA Center for Assured Software, 2017) and the NDSS18 (Le et al., 2019), can be compiled and thus enable research at the binary level. However, they are not based on real-world samples but are instead synthetic or semi-synthetic, failing to capture the complexity, diversity, and unpredictability of real-world vulnerabilities. Non-compilable datasets, such as Devign (Zhou et al., 2019), ReVeal (Chakraborty et al., 2021), BigVul (Fan et al., 2020), and others (Wang et al., 2021; Bhandari et al., 2021; Chen et al., 2023; Ding et al., 2024; Ni et al., 2024; Wang et al., 2024; He & Vechev, 2023), are mined from real-world repositories and reflect authentic vulnerability patterns, often based on security patches and developer activity. However, because they are non-compilable, they cannot be transformed into binary code, limiting their usefulness for training models that operate at the binary level.

To bridge this gap, we introduce *CompRealVul*, a novel, real-world-based compilable dataset designed to support both source and binary vulnerability detection. To build *CompRealVul*, we first constructed the CompRealVul_Raw dataset by collecting samples from ten existing datasets - Devign (Zhou et al., 2019), Big-Vul (Fan et al., 2020), ReVeal (Chakraborty et al., 2021), PatchDB (Wang et al., 2021), CVEFixes (Bhandari et al., 2021), DiverseVul (Chen et al., 2023), SEVN (He & Vechev, 2023), MegaVul (Ni et al., 2024), ReposVul (Wang et al., 2024), PrimeVul (Ding et al., 2024). which themselves were built from real-world projects, often based on security patch commits from repositories such as GitHub. As a result, CompRealVul_Raw dataset amounted to 49,832 samples, which were then processed through Compote. The samples that were successfully wrapped and

compiled formed the CompRealVul_C (C source) and CompRealVul_Bin (binary) datasets. CompRealVul_C contains labeled C functions (vulnerable = 1, non-vulnerable = 0) and CWE-ID, ready for compilation, enabling researchers to explore the effects of different compiler configurations. CompRealVul_Bin is the compiled version of CompRealVul_C, generated using `gcc` with the `-O0` optimization level. Additional details on how this dataset was created are provided in Section 5.2. We provide a comparison between existing datasets and CompRealVul in Table 1.

Table 1: Overview of vulnerability detection datasets.

| Dataset | Type | Compilable | Origin | Total Size | Vul / Non-Vul | Language |
|---------|------|-----------|--------|-----------|---------------|----------|
| Juliet (2018) Richardson (2024) NIST | Source | ✓ | Synthetic | 64,099 | 46,491 / 46,490 | C/C++ |
| NDSS18 (2019) (Le et al., 2019) | Embedding | ✓ | Semi-synth. | 28,886 8,991 | 4,490 / 4,501 | Java C |
| BinPool (2025) (Arasteh et al., 2025) | Metadata , Binary, CSV | ✓ | Real | 910 pairs | 910 / 910 | Mostly C, C++ |
| Devign (2019) (Zhou et al., 2019) | Source | ✗ | Real | 27,318 | 12,512 / 14,806 | C |
| ReVeal (2021) (Chakraborty et al., 2021) | Source | ✗ | Real | 22,740 | 2,251 / 20,489 | C |
| BigVul (2020) (Fan et al., 2020) | Source | ✗ | Real | 188,636 | 10,973 / 177,695 | C/C++ |
| PatchDB (2021) (Wang et al., 2021) | Source | ✗ | Real | 35,815 | 12,073 / 23,742 | C/C++ |
| CveFixes (2021) (Bhandari et al., 2021) | Source | ✗ | Real | 277,948 | 126,599 / 151,349 | Multi |
| DiverseVul (2023) (Chen et al., 2023) | Source | ✗ | Real | 349,437 | 18,945 / 330,492 | C/C++ |
| PrimeVul (2024) (Ding et al., 2024) | Source | ✗ | Real | 235,768 | 6,968 / 228,800 | C |
| MegaVul (2024) (Ni et al., 2024) | Source | ✗ | Real | 339,548 | 17,380 / 322,168 | C/C++ |
| ReposVul (2024) (Wang et al., 2024) | Source | ✗ | Real | 51,957 | 721 / 51,236 | C, C++, Python, Java |
| SEVN (2023) (He & Vechev, 2023) | Source | ✗ | Real | 803 pairs | 803 / 803 | C, C++, Python |
| CompRealVul (2025) | Source & Binary | ✓ | Real | C: 18,538 Binary: 18,538 | 8,141 / 10,397 | C |

## 5 EVALUATION

### 5.1 EVALUATION OF COMPOTE

To assess the capability of Compote to generate valid compilation wrappers, we used it to compile the CompRealVul_Raw dataset (described in Section 4). We implemented Compote as a Python script based on the functionality of the LangGraph package (Inc., 2024), and ran it on a 64-core AMD EPYC (3.2 GHz) CPU Linux system with 256 GB of DDR memory (no GPU was used). After running the script, we examined the results and analyzed two main factors: (i) the number of functions successfully compiled; and (ii) the number of iterations required for each test case to compile successfully. Out of 49,832 functions, Compote was able to compile 18,538 functions (forming the CompRealVul_C dataset), achieving a success rate of 37.2%. Figure 7 that in Appendix Section B.3 illustrates the distribution of successful compilations across iterations, with the maximum iteration threshold set to five. Most functions compiled on the first attempt (9,354), with a mean of 1.7 and a median of 1.0 iterations, demonstrating Compote's efficiency. These results indicate the effectiveness of Compote as a practical tool for wrapping and compiling standalone C functions.

### 5.2 EVALUATION OF COMPREALVUL_BIN DATASET IMPACT

In this section, we evaluate how the CompRealVul_Bin dataset influences the performance of binary-level vulnerability detection models. We trained six RNN-based models and two LLM-based models using four different training sets, along with two validation and test sets. These datasets consist of various subsets and combinations of two core benchmark datasets: the **Juliet Test Suite**, a

widely used synthetic dataset (from the public GitHub repository Richardson (2024)), and our newly constructed **CompRealVul_Bin**, which includes real-world vulnerabilities. Full descriptions of both core benchmark datasets are provided in Section 4. We describe this process in detail below.

**Data Preparation**. Before compilation, and to ensure compatibility between the two core benchmark datasets, we retained in Juliet only files ending with the .c extension, since **CompRealVul_Bin** contains only C functions. This involved removing all .cpp files from the **Juliet Test Suite**. After this cleanup, each benchmark dataset went through compilation. The **Juliet Test Suite** was compiled using its provided CMake configuration. Our CompRealVul_C dataset is built with Compote, which transforms each function snippet into a fully compilable program. We then compile these programs on a Linux system using GCC with the -O0 flag, producing executable binaries. Details on our compilation success rates are available in Section 5.1.

**Pre-processing**. Next, we proceeded to pre-process the resulting binary samples. This step includes two main stages: (i) converting all binary files into LLVM-IR format using the RetDec tool (Avast Software, 2024); and (ii) applying a pre-processing method based on Schaad & Binder (2023), to normalize the code and extract both vulnerable and non-vulnerable functions from the binary samples. We lift binaries to LLVM-IR because detecting vulnerabilities in raw binaries is a well-known challenge. LLVM-IR offers a more human-readable, assembly-like representation (LLVM Project) that is compiler agnostic, making it easier to generalize across different compilation settings and reduce variability from compiler-specific optimizations (McCully et al., 2024). Since lifting the binary data to LLVM-IR may create unwanted residue, the data undergoes cleaning to eliminate noise or tool-related artifacts, ensuring the resulting data is aligned for analysis (Engel et al., 2013). To maintain compatibility between the two datasets, we also removed samples where the vulnerability spanned more than a single function in the **Juliet Test Suite**. The output of this phase is a JSON file containing functions from each benchmark dataset. Every function entry includes five attributes: dataset name, file name, function name, normalized LLVM-IR function, label. The overall pre-processing phase took approximately one and a half hours for CompRealVul_Bin; and 13 hours for Juliet Test Suite to complete on a standard Linux-based machine (no GPU was used), as described in Section 5.1.

**Duplicate Elimination**. Prior research has noted the presence of duplicates in benchmarks derived from the Juliet Test Suite (Brust et al., 2023). While each Juliet test case starts out as unique, duplicates tend to emerge during subsequent steps, such as extraction, pre-processing, or compilation using various optimizations (Brust et al., 2023). In some cases, more than 90% of binary-level samples produced from the Juliet dataset were identified as duplicates, emphasizing the challenges of using synthetic benchmarks (Russell et al., 2018). To address this concern, following the pre-processing, we implemented a duplicate elimination mechanism. Overall, we found that 74.3% of the pre-processed Juliet dataset consisted of duplicate entries, totaling 44,258 out of 59,569 files. In comparison, in CompRealVul_Bin, 1,270 out of 18,538 samples were identified as duplicates.

**Sequence Length Filtering.** After generating the final JSON files, we examined the length distribution of the functions to determine a suitable truncation threshold. As shown in Figure 8 (at the Appendix), only 706 functions exceed 4,096 tokens while using ModernBert-large (Warner et al., 2024) tokenizer. Based on this observation, we set the maximum sequence length to 4,096 for all models trained with this data; and excluded samples which exceeded this length during the training phase.

**Datasets splits**. The data was split into several training, validation, and test sets (see Table 2).

### 5.2.1 EXPERIMENTAL SETUP & EVALUATION CRITERIA

We evaluated our models under four distinct train–test regimes that cover both real-synthetic comparisons and a combined setting. Specifically, we trained on CompRealVul and tested on CompRealVul to measure performance on real-world binaries; trained on Juliet and tested on Juliet to capture performance on synthetic data; trained on a Juliet subset and tested on CompRealVul to assess synthetic-real generalization; and finally trained on the combined dataset and tested on CompRealVul to examine mixing synthetic and real data. Across these four experiments, each architecture was trained or fine-tuned, resulting in 32 total model runs. Specifically, we fine-tune two distinct LLM architectures: ModernBERT-large (Warner et al., 2024) (encoder-only) and StarCoder2-3B (Lozhkov et al., 2024) (decoder-only). We also train two different architectures of simple RNN, two different architectures of LSTM, and two different architectures of GRU architectures; inspired by the method-

Table 2: Overview of training, validation, and testing splits for CompRealVul and Juliet-based datasets. Vul=Vulnerability, non-Vul=non-vulnerability

| Set Name | Vul/non-Vul (Total) | Explanation |
|---|---|---|
| **Training Sets** | | |
| CompRealVul_train | 5,928/6,156 (12,084) | Training set was created by splitting the pre-processed CompRealVul_bin dataset into 70% train, 10% validation, and 20% test, while maintaining a vulnerable to non-vulnerable ratio of 1:3 in validation and testing sets. This is the 70% train part. |
| Juliet_Regular_train | 4,089/6,628 (10,717) | A random split of Juliet after pre-processing. It assigns 70% for training, 10% for validation, and 20% for testing, ensuring vulnerable samples appear in all subsets. This is the 70% train part. |
| Juliet_CompRealVul_train | 5,831/6,253 (12,084) | This training set was constructed from Juliet to maintain a fair comparison with CompRealVul_train. This dataset satisfies the following constraints: (i) it has the same total number of samples (12,084) as CompRealVul_train; and (ii) it maintains a similar vulnerable/non-vulnerable sample ratio. Since Juliet has only 5,831 vulnerable samples after pre-processing, we used all of them and added enough non-vulnerable samples to reach the target size. |
| Combined_train | 11,759/12,409 (24,168) | This training set is the union of the CompRealVul_train and Juliet_CompRealVul_train sets. |
| **Validation Sets** | | |
| CompRealVul_val | 432/1,296 (1,728) | Validation set was created by splitting the pre-processed CompRealVul_bin dataset into 70% train, 10% validation, and 20% test, maintaining a vulnerable to non-vulnerable ratio of 1:3 in validation and testing sets. This is the 10% validation part. |
| Juliet_Regular_val | 579/952 (1,531) | A random split of Juliet after pre-processing. It assigns 70% for training, 10% for validation, and 20% for testing, ensuring vulnerable samples are present in all subsets. |
| **Testing Sets** | | |
| CompRealVul_test | 864/2,592 (3,456) | The test set was created by splitting the pre-processed CompRealVul_bin dataset into 70% train, 10% validation, and 20% test, maintaining a vulnerable to non-vulnerable ratio of 1:3 in validation and testing sets. This is the 20% test part. |
| Juliet_Regular_test | 1,900/1,163 (3,063) | A random split of Juliet after pre-processing. It assigns 70% for training, 10% for validation, and 20% for testing, ensuring vulnerable samples are present in all subsets. This is the 20% test. |

ology described in Schaad & Binder (2022). In contrast, we used longformer-base-4096 (Beltagy et al., 2020) for embedding the inputs. More details on the architectures and the fine-tuning/training parameters, memory usage, and run-time are provided in Table 5 and Table 6 in Appendix Section B.2. By leveraging 8 distinct architectures, we evaluate how different model types perform with these datasets. All training and fine-tuning were conducted on a Linux-based system with a 64-core AMD EPYC (3.2 GHz) CPU and 256 GB of DDR memory, equipped with an NVIDIA RTX A6000 GPU.

To evaluate model performance, we use the traditional metrics of accuracy, precision, recall, and F1 score. We also incorporate the Vulnerability Detection Score (VD-S) metric introduced in Ding et al. (2024), which specifically measures the false negative rate after the detector is calibrated to maintain a false positive rate below a certain threshold (15% in our case).

### 5.2.2 EXPERIMENTAL RESULTS

We fine-tuned or trained (as applicable) and evaluated each model for each training set shown in Table 2. This resulted in four different experiments per model - each for a training set. Table 3 and Table 4 present the evaluation results of RNN-based models and LLMs, respectively. Starting with the RNN-based models in Table 3, we observe that models trained exclusively on the Juliet benchmark (i.e., Juliet_Regular_train), a synthetic dataset, perform well only when tested on Juliet itself. Their metrics drop sharply when tested on vulnerabilities sourced from real-world data, as represented by CompRealVul_Bin. For example, the BiLSTM (2 layers) model shows a dramatic increase in VD-S, from 21.75% (Juliet test) to 86.41% (CompRealVul_Bin test), demonstrating the inability of synthetic-only training data to generalize to realistic scenarios.

Note, however, that even when models are trained and tested exclusively on CompRealVul_Bin, they struggle to achieve better performance. While they perform better than models trained solely on Juliet (as reflected by substantial differences in loss values), the modest accuracy and F1 scores suggest that these RNN and LLM-based models still struggle to adequately learn the complex patterns of real-world binary vulnerabilities. This may indicate that the intricacies of such vulnerabilities are too complex for standard RNN architectures, and perhaps even for fine-tuned LLMs, to capture effectively, highlighting the need for new, specialized architectures to address this problem.

We also note that a partial relief to this issue was achieved by training on a combined dataset of Juliet and CompRealVul_Bin (i.e., Combined_train). This approach consistently improved precision, F1-score, and AUC when testing for real world vulnerabilities (as they manifest in CompRealVul_test),

while maintaining more balanced VD-S scores compared to Juliet-only training. These results demonstrate that combining real and synthetic data leads to models that are more robust in dealing with real-world cases. Synthetic to real generalization gap phenomena were also observed in Table 4, which provides details on the evaluation of LLMs - ModernBERT and StarCoder. ModernBERT achieves extremely high results on the Juliet dataset (e.g., 99.22% accuracy, 0% VD-S), but, like the RNN models, struggles when applied to real-world data. Training on the Combined_train dataset mitigates this drop, boosting all metrics, and confirming that the dataset we propose contributes to better generalization even for large pre-trained models.

Table 3: Performance of RNN-based models on various train-test combinations.

| Model | # Layers | Train | Test | Loss | Accuracy | Precision | Recall | F1 | VD-S | AUC |
|-------|----------|-------|------|------|----------|-----------|--------|-----|------|-----|
| BiGRU | 1 layer | CompRealVul_train | CompRealVul_test | 0.71352 | 24.72% | 24.01% | 98.88% | 38.64% | 81.42% | 52.73% |
| | | Combined_train | CompRealVul_test | 0.67956 | 55.35% | 24.33% | 40.90% | 30.51% | 82.92% | 51.12% |
| | | Juliet_CompRealVul_train | CompRealVul_test | 0.85349 | 51.49% | 22.50% | 41.90% | 29.28% | 83.29% | 48.04% |
| | | Juliet_Regular_train | Juliet_Regular_test | 0.44730 | 78.45% | 67.53% | 83.32% | 74.60% | 24.42% | 88.87% |
| | 2 layers | CompRealVul_train | CompRealVul_test | 0.69852 | 39.42% | 24.10% | 71.07% | 35.99% | 84.66% | 51.08% |
| | | Combined_train | CompRealVul_test | 0.76395 | 36.31% | 23.85% | 75.56% | 36.25% | 84.53% | 49.96% |
| | | Juliet_CompRealVul_train | CompRealVul_test | 0.78228 | 50.39% | 23.22% | 46.38% | 30.95% | 85.04% | 48.90% |
| | | Juliet_Regular_train | Juliet_Regular_test | 0.39970 | 82.11% | 77.43% | 74.63% | 76.01% | 22.36% | 89.37% |
| BiLSTM | 1 layer | CompRealVul_train | CompRealVul_Bin | 0.72001 | 23.97% | 23.97% | 100.00% | 38.67% | 81.55% | 52.42% |
| | | Combined_train | CompRealVul_test | 0.71280 | 25.16% | 23.99% | 97.88% | 38.54% | 82.79% | 51.10% |
| | | Juliet_CompRealVul_train | CompRealVul_test | 0.86882 | 46.89% | 23.49% | 53.87% | 32.71% | 83.92% | 49.77% |
| | | Juliet_Regular_train | Juliet_Regular_test | 0.36560 | 83.19% | 77.46% | 78.59% | 78.02% | 20.46% | 91.27% |
| | 2 layers | CompRealVul_train | CompRealVul_test | 0.67822 | 59.80% | 24.27% | 31.92% | 27.57% | 84.66% | 50.99% |
| | | Combined_train | CompRealVul_test | 0.72102 | 34.13% | 23.65% | 78.43% | 36.34% | 83.67% | 50.11% |
| | | Juliet_CompRealVul_train | CompRealVul_test | 0.71521 | 23.97% | 23.97% | 100.00% | 38.67% | 86.41% | 48.71% |
| | | Juliet_Regular_train | Juliet_Regular_test | 0.37485 | 82.47% | 76.26% | 78.16% | 77.20% | 21.75% | 90.83% |
| BiRNN | 1 layer | CompRealVul_train | CompRealVul_test | 0.70200 | 41.00% | 24.60% | 70.70% | 36.50% | 82.40% | 52.50% |
| | | Combined_train | CompRealVul_test | 0.70800 | 32.30% | 24.30% | 86.20% | 37.90% | 81.30% | 52.00% |
| | | Juliet_CompRealVul_train | CompRealVul_test | 0.69200 | 52.40% | 25.80% | 52.60% | 34.60% | 83.00% | 52.20% |
| | | Juliet_Regular_train | Juliet_Regular_test | 0.49700 | 75.40% | 66.90% | 69.60% | 68.20% | 40.50% | 82.70% |
| | 2 layers | CompRealVul_train | CompRealVul_test | 0.68300 | 61.50% | 26.20% | 33.40% | 29.40% | 82.30% | 52.40% |
| | | Combined_train | CompRealVul_test | 0.71100 | 47.50% | 24.70% | 58.00% | 34.60% | 82.40% | 52.10% |
| | | Juliet_CompRealVul_train | CompRealVul_test | 0.78000 | 45.80% | 23.90% | 57.50% | 33.70% | 84.20% | 50.60% |
| | | Juliet_Regular_train | Juliet_Regular_test | 0.52500 | 72.80% | 60.40% | 82.40% | 69.70% | 39.60% | 83.20% |

Table 4: Performance of pre-trained models (ModernBERT, StarCoder) on various train-test splits.

| Model | Train | Test | Loss | Accuracy | Precision | Recall | F1 | VD-S | AUC |
|-------|-------|------|------|----------|-----------|--------|-----|------|-----|
| ModernBERT | CompRealVul_train | CompRealVul_test | 0.73707 | 55.68% | 23.61% | 37.89% | 29.09% | 82.36% | 50.87% |
| | Combined_train | CompRealVul_test | 0.82691 | 40.48% | 24.62% | 71.80% | 36.66% | 81.74% | 52.29% |
| | Juliet_CompRealVul_train | CompRealVul_test | 6.51893 | 38.51% | 23.70% | 70.43% | 35.47% | 84.72% | 49.60% |
| | Juliet_Regular_train | Juliet_Regular_test | 0.07910 | 99.22% | 98.97% | 98.97% | 98.97% | 0.00% | 99.91% |
| StarCoder | CompRealVul_train | CompRealVul_test | 0.73220 | 45.78% | 24.60% | 61.85% | 35.20% | 84.41% | 50.65% |
| | Combined_train | CompRealVul_test | 1.6977 | 44.16% | 23.38% | 60.05% | 33.66% | 84.79% | 50.27% |
| | Juliet_CompRealVul_train | CompRealVul_test | 12.219 | 38.21% | 23.63% | 72.55% | 35.64% | 100.00% | 50.18% |
| | Juliet_Regular_train | Juliet_Regular_test | 0.2338 | 96.57% | 96.40% | 94.50% | 95.44% | 0.02% | 99.50% |

# 6 CONCLUSION AND FUTURE WORK

We introduced Compote, a compilation wrapper tailored for standalone C code snippets, and released two datasets: CompRealVul_C and CompRealVul_Bin. Our experimental results demonstrate that binary-level vulnerability detection models trained solely on synthetic datasets struggle to generalize to real-world scenarios. By augmenting existing datasets with CompRealVul_Bin, we show an improvement in the ability of vulnerability detection models to handle more realistic and complex code. While our findings suggest that combining synthetic and real-world data improves model performance, current vulnerability detection models still fall short in fully capturing the complex patterns and subtle semantics of real-world vulnerabilities. This limitation indicates that, although datasets like CompRealVul_C, that can be compiled, bring us closer to realistic evaluation settings, the models themselves still require further refinement. Several limitations of our approach should be acknowledged. First, Compote was applied only to functions shorter than 2,500 characters, as longer functions were more difficult for Compote to wrap successfully. Second, during the automated decompilation and LLVM lifting process with RetDec, we observed unintended over-optimization, similar to what was reported in Schaad & Binder (2022). This behavior can simplify the code to the point that certain functions and potentially even vulnerabilities are omitted from the resulting LLVM-IR representation of the binary. Future work can build on the CompRealVul_C dataset to enhance model robustness and use CompRealVul_Bin as a real-world evaluation benchmark.

## 7 REPRODUCIBILITY

Upon acceptance, we will release both the code and the dataset.

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

## A  USE OF LARGE LANGUAGE MODELS (LLMs)

In accordance with the ICLR 2026 policy on large language model (LLM) usage, we disclose that LLMs were used as assistive tools in this research. They supported coding (writing and fixing scripts), writing and analysis (improving the text, structure, and interpretation of results), and knowledge retrieval (related work and information about existing LLMs). All outputs from LLMs were carefully checked and edited if needed by the authors, who take full responsibility for the final paper.

## B  TECHNICAL APPENDICES AND SUPPLEMENTARY MATERIAL

The following list provides a structured overview of the appendix contents, indicating what each section contains and referring to relevant figures and tables.

- **Compote Prompts & Algorithm ( Section B.1)**
  See  Section 3 for further explanation of the related figures.

  - Figure 2: Prompt template used during the Initial Code Generation phase.
  - Figure 3: Prompt template used during the Error-Based Revision phase.
  - Figure 4: Prompt template for injecting a function call into `main`.
  - Figure 5: Pseudo-code illustrating the iterative workflow of the compilation agent.
  - Figure 6: Wrapping a Standalone C Function into a Compilable Program with Compote.

- **Tables ( Section B.2)**
  See  Section 5 for further explanation of the related tables.

  - Table 5: Fine-tuning hyperparameters for transformer models.
  - Table 6: Fine-tuning hyperparameters for the RNN models.

- **Evaluation Figures ( Section B.3)**
  See  Section 5 for further explanation of the related figures.

  - Figure 7: Histogram of success iterations per sample.
  - Figure 8: Function length distribution across datasets.

## B.1 COMPOTE PROMPTS & ALGORITHM

```
## Context
You are an expert in compilation of C code. You have been asked to add
code to the following standalone C function so that it compiles
successfully.
Your task is to transform this standalone C function into a complete,
compilable C program.

## Instructions
1. Complete the provided C code to ensure it compiles successfully
   by adding necessary headers, a main function, and any stubs for
   called functions.
2. Do not alter the input function at all.

## Output Format
- The result should be pure C code, suitable for compilation
  without any additional text, comments, or explanation.

## Input Function:
{input_function}
```

Figure 2: Prompt template used during the Initial Code Generation phase. The LLM is instructed to wrap a given C function in a complete, compilable program without altering the original function.

```
## Context
We have a C code snippet that fails to compile.
We want to correct the compilation errors while
preserving the existing function and its signature.

## Instructions
1. Review the compilation errors provided:
{compilation_errors}
2. Fix the code so it compiles successfully.
3. Do not modify the {function_signature} function content
   or its signature.
4. Add only the necessary:
     - Include headers
     - main() function
     - Stub functions (if any are called but not defined)
5. Do not include any additional text, comments,
   or explanations in your response.

## Output Format
Return pure C code in a single block, suitable
for compilation without further modifications.

## Input C code:
{input_function}
```

Figure 3: Prompt template used during the Error-Based Revision phase. The LLM is instructed to revise previously generated C code using compiler error messages while preserving the original function and its signature.

```
## Task
Modify the `main` function in the provided C code to add a valid function call to `{
    function_name}`.

## Target Function Signature
```c
{cleaned_signature}
```

## Instructions
1.  Locate/Create `main`: Find the `main` function. If it doesn't exist, create a
    basic one (`int main(...) {{ /* Call here */ return 0; }}`).
2.  Add Call: Inside `main`'s body, add *one* function call to `{function_name}`.
3.  CRITICAL - Arguments: Generate valid arguments strictly based on the *Target
    Function Signature*. Declare and initialize necessary local variables *within `
    main`* just before the call. Ensure the arguments allow the code to compile.
4.  CRITICAL - Preservation: Modify *only* the `main` function as needed to add the
    call and its argument variables. DO NOT alter the target function's definition,
    other existing code in `main`, includes, globals, or any other part of the file.
5.  Output Only Code: Return *only* the complete, modified C code block, ready for
    compilation. No extra text, comments, or explanations.

## Input C code:
{input_function}
```

Figure 4: Prompt template used during the Function Call Insertion phase. The LLM is instructed to modify or create a `main` function that calls the specified target function using valid arguments based on its signature, without altering any other part of the code.

```
Pseudo-code for the Compote workflow:

Input:
    F = { f_1, f_2, ..., f_n }   (a set of standalone C functions)
    m                            (maximum number of iterations)

For each function f_i in F:
    iteration <- 0
    f_name <- func_name(f_i)
    code <- LLM_Generate(f_i)    # Code Wrapping

    while iteration < m:
        code <- OverwriteWithOriginalFunction(code, f_i)
        (success, errors) <- Compile(code)
        if success:
            is_function_call <- IsFunctionCallInMain(code, f_name)
            if is_function_call:
                SaveSample(code)
                break
            else:
                code <- AddFunctionCallInMain(code, f_name)
                iteration <- iteration + 1
        else:
            code <- LLM_Revise(code, errors)
            iteration <- iteration + 1

    if (iteration = m) and (success = false):
        # Log failure for f_i
```

Figure 5: Illustration of the iterative process for code generation, compilation, and error-driven revision.

```
void iwjson_ftoa(long double val, char buf[static IWNUMBUF_SIZE], size_t *out_len) {
  int len = snprintf(buf, IWNUMBUF_SIZE, "%.8Lf", val);
  if (len <= 0) {
    buf[0] = '\0';
    *out_len = 0;
    return; }
  while (len > 0 && buf[len - 1] == '0') {
    buf[len - 1] = '\0';
    len--; }
  if ((len > 0) && (buf[len - 1] == '.')) {
    buf[len - 1] = '\0';
    len--; }
  *out_len = (size_t) len; }
```

```
#include <stdio.h>
#include <stddef.h>

#define IWNUMBUF_SIZE 32

void iwjson_ftoa(long double val, char buf[static IWNUMBUF_SIZE], size_t *out_len) {
  int len = snprintf(buf, IWNUMBUF_SIZE, "%.8Lf", val);
  if (len <= 0) {
    buf[0] = '\0';
    *out_len = 0;
    return; }
  while (len > 0 && buf[len - 1] == '0') {
    buf[len - 1] = '\0';
    len--; }
  if ((len > 0) && (buf[len - 1] == '.')) {
    buf[len - 1] = '\0';
    len--; }
  *out_len = (size_t) len; }

int main() {
    long double value = 123.45678901;
    char buffer[IWNUMBUF_SIZE];
    size_t length;

    iwjson_ftoa(value, buffer, &length);
    printf("Converted value: %s (length: %zu)\n", buffer, length);

    return 0;
}
```

Figure 6: Wrapping a Standalone C Function into a Compilable Program with Compote (top: original function, bottom: wrapped compilable program)

## B.2 TABLES

Table 5: Fine-tuning hyperparameters for transformer models
** Time per epoch for longest case

| Model | Type | Params | Batch Size | LR | Epochs | LoRA Config | GPU / Time Per Epoch |
|---|---|---|---|---|---|---|---|
| ModernBERT-large | Encoder | 395M | 4 | 2e-5 | 20 | – | 40.42 GB / 3h |
| StarCoder | Decoder | 3B | 2 | 2e-5 | 20 | Rank=16, $\alpha$=32, Dropout=0.05 | 35.45 GB / 6h |

Table 6: Training hyper-parameters for models
** Time per epoch for longest case

| Model | Layers | Unit Size | Batch Size | LR | Epochs | GPU / Time Per Epoch |
|---|---|---|---|---|---|---|
| BiGRU | 1 | 128 | 32 | 2e-5 | 50 | 26.16 GB / 1.5h |
| | 2 | 128 | 32 | 2e-5 | 50 | 26.17 GB / 1.5h |
| BiRNN | 1 | 128 | 32 | 2e-5 | 50 | 26.16 GB / 1.5h |
| | 2 | 128 | 32 | 2e-5 | 50 | 30.27 GB / 1.5h |
| BiLSTM | 1 | 128 | 32 | 2e-5 | 50 | 26.67 GB / 1.5h |
| | 2 | 128 | 32 | 2e-5 | 50 | 26.68 GB / 1.5h |

## B.3    EVALUATION FIGURES

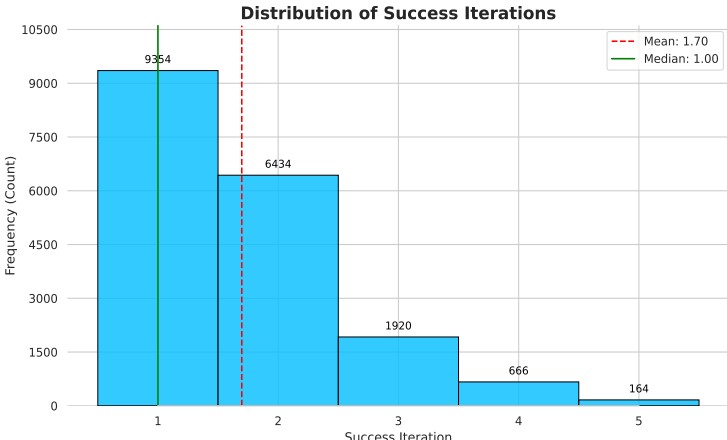

Figure 7: Histogram of compilation success.

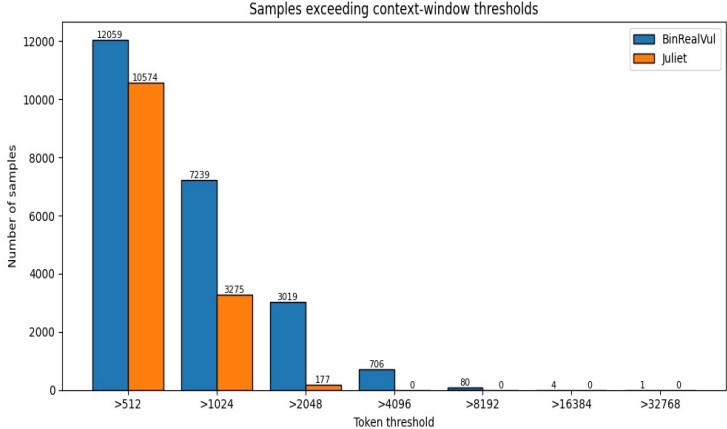

Figure 8: Function length distribution across both datasets.

