# OpenReview forum: "COMPOTE: Generating a Dataset of Real-World Binary Level Vulnerabilities"
_ICLR.cc/2026/Conference — ICLR 2026 Conference Desk Rejected Submission_

### Official Review · Reviewer_C4Tu · 2025-10-26

**Soundness:** 2
**Presentation:** 2
**Contribution:** 2
**Rating:** 4
**Confidence:** 3

**Summary:**

This paper proposes the Compote tool and CompRealVul dataset to address the lack of high-quality real-world data in binary vulnerability detection. Through AI-driven code completion technology, Compote can automatically convert non-compilable C code fragments into compilable programs and generate corresponding binary files. This work not only fills the gap in real benchmark data for binary vulnerability detection but also reveals the limitations of current detection technologies in real scenarios, providing important datasets and methodological references for future research.

**Strengths:**

The paper's core contribution lies in filling the long-standing gap of real-world data in binary vulnerability detection. Through innovative AI-assisted compilation technology (Compote), it transforms fragmented source code vulnerability datasets into compilable, executable binary samples. This not only solves the practical engineering challenge of "source code unavailability, only binaries available" for detection, but also rigorously reveals the generalization gap between synthetic data and real vulnerabilities through experiments. It provides data support and methodological inspiration for subsequent research, demonstrating significant practical value and academic significance.

**Weaknesses:**

The study presents several critical issues requiring further clarification and improvement. Primarily, while the authors innovatively proposed the Compote tool for processing non-compilable code, there lacks systematic review of existing similar work. In binary vulnerability research, have other teams attempted alternative approaches (e.g., manual completion or semi-automatic conversion) for such data? This necessitates citation and comparative analysis of relevant literature. Secondly, the code completion process may alter the triggering conditions of original vulnerabilities, particularly for multi-threaded or memory-operation vulnerabilities relying on complex contexts. The authors should provide more rigorous verification methods (e.g., dynamic testing or symbolic execution) to demonstrate the semantic integrity of vulnerabilities in completed code. Thirdly, regarding dataset construction, CompRealVul essentially derives from existing datasets, with its innovation mainly reflected in the compilation toolchain rather than the data per se. This "reprocessing" paradigm requires clearer positioning in academic contribution evaluation. Furthermore, the experimental design shows notable shortcomings: horizontal comparisons are limited to internal models, failing to benchmark against state-of-the-art code completion systems (e.g., GitHub Copilot or CodeLlama); the baseline models also exclude newer architectures proposed in 2023-2024. Most crucially, certain experimental results (e.g., accuracy degradation) contradict the paper's central claims, demanding deeper analysis to explain these discrepancies. We recommend supplementing more comprehensive comparative experiments and robustness tests to strengthen the conclusions. The authors should particularly address how their method maintains vulnerability authenticity during code completion, and how the compiled binaries compare with manually verified ground truth in terms of vulnerability preservation.

**Questions:**

1.	The authors mention that existing real datasets are difficult to directly compile into binary files. Have there been previous attempts to reprocess these datasets to generate binary versions? If so, references and comparisons are needed.
2.	When completing code, how to avoid changing the trigger conditions of original vulnerabilities due to automatically generated content? Could this potentially transform complex vulnerabilities into simpler ones?
3.	The CompRealVul_C and CompRealVul_Bin datasets are essentially derived from existing datasets (e.g., Devign) rather than originally collected. The authors should clarify that their contribution is the "compilation toolchain" rather than the "data itself," and discuss the academic value and innovation of such derived data.
4.	Many code completion models (e.g., GitHub Copilot) already exist. The authors should compare Compote's performance in "compilation-oriented completion" with these models to demonstrate its irreplaceability. Current experiments only show internal comparisons, lacking horizontal references.
5.	In model comparison experiments, were all models tested with identical hyperparameters, hardware environments, and training durations? If data scales differ (e.g., Juliet's sample size is much larger than CompRealVul's), the authors should explain whether sampling balance was implemented, as this may affect conclusion credibility.
6.	Are the baseline models (e.g., BiLSTM) the most advanced in this field? For example, have better architectures been proposed recently (e.g., in 2024)? Additionally, comparisons with commercial large models (e.g., GPT-4, Cursor), which excel in code understanding tasks, are missing. The authors should explain why these weren't included.
7.	Code added for compilation may interfere with models' understanding of original function logic, especially for long code segments. The authors should discuss whether this affects models' focus on core vulnerabilities.
8.	Some metrics (e.g., Accuracy, Precision) show that models trained on CompRealVul still underperform those trained on Juliet, contradicting the hypothesis that "real data improves generalization." This makes it difficult to demonstrate the dataset's advantages.

---

> ### Author Response · Authors · 2025-11-21
>
> **Response for question 1**
>
> To the best of our knowledge, none of the real-world vulnerability datasets we compile in
> this paper have ever been compiled into large-scale binary-level datasets in prior work. All of
> these benchmarks are released at the source-code level, and no published system provides a
> general, reusable compilation pipeline that can process all their functions. From the system-
> methodology perspective, as cited in our related work section, Le et al. (2019) [1] proposed
> a deterministic, rule-based method for repairing incomplete code snippets so they could be
> compiled into binary functions. To the best of our knowledge, their tool was not publicly
> released, so we were not able to compare COMPOTE directly with their system. COMPOTE
> differs in both purpose and methodology. Our contribution is an automated, LLM-guided
> compilation flow that transforms real-world vulnerable functions into compilable standalone
> programs while ensuring the original function remains unchanged. This enables the creation
> of a binary-level vulnerability dataset at scale.
> [1] Tue Le, Tuan Vu Nguyen, Trung Le, Dinh Phung, Paul Montague, Olivier De Vel, and
> Lizhen Qu. Maximal divergence sequential auto-encoder for binary software vulnerability
> detection. In International Conference on Learning Representations 2019. International
> Conference on Learning Representations (ICLR), 2019
>
> **Response for question 2**
>
> In COMPOTE, the original function is never changed. The LLM only adds minimal
> scaffolding such as headers or small stubs to make the code compilable. Since the vulnerable
> function itself remains exactly as in the source dataset, the triggering conditions of the
> vulnerability are preserved.
>
> **Response for question 3**
>
> The core contribution is the creation of a capability to generate binary-level vulnerability
> datasets from real-world vulnerable functions, as we discuss in detail in the Introduction
> section: "To address this challenge, we introduce Compote, an AI-based tool designed to
> automatically transform standalone C functions into fully compilable programs. [...] In the
> context of binary-level vulnerability detection, the availability of suitable training datasets
> is severely limited: currently, only three main publicly available datasets exist [...] The
> source code level vulnerability detectors benefit from a plethora of high-quality datasets
> derived from real-world projects [...] While these datasets provide a realistic foundation
> for vulnerability detection research, they cannot be directly compiled, since they contain
> partial code snippets rather than complete programs that meet compilation requirements.
> This limitation prevents the use of these datasets to train binary-level vulnerability detectors.
> Compiling these datasets would create binary-level datasets, alleviating the scarcity of
> training sets at the binary level and potentially improving binary-level vulnerability detectors.
> To bridge this gap between high-quality source-level vulnerability datasets and binary-
> level vulnerability detection, we applied Compote to process existing source-level datasets,
> thereby generating two new datasets [...]." The primary contribution is a practical benchmark
> for the binary domain, and COMPOTE serves as the enabling infrastructure for producing
> standardized, real-world binary datasets going forward. This contribution aligns directly with
> the conference call for papers under application-driven machine learning, which explicitly
> values practical, innovative datasets that address real-world needs. In this spirit, we release
> a unique, standardized, and application-driven binary dataset that the community could not
> previously access.
>
> **Response for question 4**
>
> Our goal in this work is not to introduce a new code-completion model, but rather a model-
> agnostic compilation workflow that transforms existing source-level vulnerability datasets
> into compilable binaries while keeping the original vulnerable function unchanged. In
> COMPOTE, the LLM is only one part of a constrained pipeline (wrapping + compile-
> feedback revision), and in principle any sufficiently capable LLM could be used in this
> role. A direct comparison with proprietary tools such as GitHub Copilot is also difficult in
> practice, given licensing restrictions, limited batch APIs, and the challenge of reproducing
> such results within the research community.

---

> > ### Author Response · Authors · 2025-11-21
> >
> > **Response for question 5**
> >
> > Yes. Every model architecture across all experiments was trained under identical conditions,
> > with the same hardware, training duration, and optimization (Tables 5 and 6 in appendix).
> > To ensure fairness, we aligned the sizes and class ratios when comparing the Juliet and
> > CompRealVul subset, as described in the paper. This prevents dataset size from influencing
> > conclusions and allows the evaluation to isolate the effect of real-world complexity rather
> > than differences in scale.
> >
> > **Response for question 6**
> >
> > In this work, the models are not the novelty and not the main focus. Our primary contribution
> > is the release of a new, realistic binary-level vulnerability dataset. We therefore chose to
> > evaluate CompRealVul using standard and widely adopted baselines in ML4VD research,
> > including those used in Schaad & Binder (2023). In ML4VD research, a state-of-the-
> > art practice is to employ RNN-based architectures (RNN, LSTM, GRU) for binary-level
> > vulnerability detection. In addition to these classical baselines, our evaluation also includes
> > two modern LLM architectures- ModernBERT-large and StarCoder2-3B, providing coverage
> > of both traditional sequence models and contemporary transformer-based models.
> >
> > **Response for question 7**
> >
> > The added code for compilation does not interfere with the model’s understanding of the
> > original function. (1) The vulnerable function itself is never modified; COMPOTE always
> > overwrites the function body with the original version to guarantee function integrity. (2)
> > The additional code serves only as a minimal wrapper to enable compilation, which was not
> > possible beforehand. (3) During preprocessing for training and evaluation, we extract from
> > the LLVM-IR file only the function, so the model is exposed solely to the function itself.
> > Together, these steps ensure that the model remains focused exclusively on the original
> > vulnerable logic.
> >
> > **Response for question 8**
> >
> > We would like to clarify that our main hypothesis is not that training on real data will
> > automatically yield higher Accuracy or Precision than training on Juliet. Rather, our
> > central claim is that evaluating solely on synthetic datasets like Juliet can give an inflated
> > impression of model capability, whereas real-world data exposes the current limitations of
> > ML-based vulnerability detection. In this light, the fact that models trained and evaluated
> > on CompRealVul achieve performance that is similar to those trained on Juliet does not
> > contradict our hypothesis-it supports it, by showing that current models struggle once
> > realistic complexity is introduced. This underscores both the limitations of existing ML
> > models for binary vulnerability detection and the need for better, more realistic benchmarks
> > so that future ML systems can be evaluated on reliable data and ultimately improve.
> > [1] Y. Ding et al., "Vulnerability Detection with Code Language Models: How Far are We?,"
> > in 2025 IEEE/ACM 47th International Conference on Software Engineering (ICSE), Ottawa,
> > ON, Canada, 2025, pp. 1729-1741, doi: 10.1109/ICSE55347.2025.00038.

---

### Official Review · Reviewer_raRx · 2025-11-01

**Soundness:** 3
**Presentation:** 3
**Contribution:** 3
**Rating:** 8
**Confidence:** 3

**Summary:**

This paper introduces a new benchmark that includes binary level vulnerabilities. It is based on existin source code vulnerability bechmarks were source code samples are automatically completed to generate compilable code.

**Strengths:**

Vulnerability detection at the binary level is a challenging problem. Thos new benchmark promises to generate new research in this domain. The benchmark seems well curated and the methods to compile the samples appear sound.

**Weaknesses:**

The benchmark is limitted by existing source code datasets which themselves are limited and may have errors.

**Questions:**

How do you envision applying Compote to repository-level vulnerabilities?

---

> ### Author Response · Authors · 2025-11-21
>
> **Response for weakness 1**
>
> Thank you for highlighting this point. We agree that current source-level vulnerability
> datasets have inherent limitations. COMPOTE does not alter or reinterpret their labels;
> instead, it provides a systematic way to extend any such dataset into the binary domain while
> preserving the original function exactly. The benchmark we present is only one instantiation.
> COMPOTE itself is not restricted to the datasets used in the paper and can be applied to any
> existing or future source-level vulnerability collection. This makes the framework flexible,
> forward-compatible, and capable of supporting the creation of many additional real-world
> binary datasets as the field progresses.
>
> **Response for question 1**
>
> Thank you for the thoughtful question. At this stage, COMPOTE is intentionally focused on function-
> level samples, because our goal was to make existing vulnerability benchmarks compilable
> without changing their original structure or assumptions. Repository-level vulnerabilities
> involve broader context- multiple files, build systems, and interprocedural interactions,
> which we did not aim to reconstruct in this work. That said, we see COMPOTE as a first
> step toward richer, more realistic binary datasets. While extending the approach to full
> repositories would require solving challenges beyond the scope of this paper, we think the
> idea is promising and could be explored in future work.

---

> > ### Comment · Reviewer_raRx · 2025-11-23
> >
> > Thanks for the response. I remain positive.

---

### Official Review · Reviewer_VABE · 2025-11-01

**Soundness:** 2
**Presentation:** 2
**Contribution:** 2
**Rating:** 4
**Confidence:** 4

**Summary:**

The paper proposed COMPOTE, an LLM-based to automatically transform standalone C functions into fully compilable programs, which was applied to real-world functions from ten public datasets of vulnerable code, yielding a dataset comprising 18K compilable C functions along with their compiled binary versions.

**Strengths:**

**Originality**
The paper proposed novel approach of scaffolding source-code vulnerability dataset to construct compilable binary vulnerability dataset.

**Quality**
The paper made extensive experiments on various DL models and LLMs, showing quantitative results of vulnerability detection accuracy.

**Significance**
The paper proposed a framework which can be used to help further research in the field of machine learning for vulnerability detection.

**Weaknesses:**

**Flawed Problem Setting of Function-Level Vulnerability Detection**
Recent works have criticized the common practice of treating machine learning for vulnerability detection (ML4VD) as function-level binary-classification problem. As the vulnerability always depends on external context, it's unreasonable to decide if a single function is vulnerable or not. The popular datasets like Devign/BigVul/DisverseVul have problems in the quality of vulnerability labelling. The paper's approach of scaffolding vulnerable/benign functions, despite keeping the same semantic of the given function, may still change the existence of vulnerability in the context of whole program and compiled binary. Without addressing this risk, the quantitative results of the ML4VD score are not reliable.
- On the Effectiveness of Function-Level Vulnerability Detectors for Inter-Procedural Vulnerabilities (ICSE '24)
- Top Score on the Wrong Exam: On Benchmarking in Machine Learning for Vulnerability Detection (ISSTA '25)
- It Only Gets Worse: Revisiting DL-Based Vulnerability Detectors from a Practical Perspective (APSEC '25)

**Lack of Qualitative Example of Scaffolding**
Scaffolding arbitrary code from real-world software is a non-trivial task which requires correct API and initialization setup. Even if the resulting program is compilable, it does not mean the code is functionally correct. It would be better if the authors can provide at least some example of vulnerable code snippets and resulting compilable code.

**Questions:**

1. What real-world software does the dataset include?
2. Does the scaffolding consider semantic correctness beyond syntactic compilability?

---

> ### Author Response · Authors · 2025-11-21
>
> **Response for weakness 1**
>
> Thank you for raising this point. We acknowledge recent work discussing the limitations
> of function-level vulnerability detection. Our goal is not to claim that all vulnerability
> types can be captured at this granularity, but to support research in a domain where a large
> portion of the binary-level ML literature still operates. Function-level detection is a well-
> established and actively studied setting in both source-code and binary analysis, and many
> widely used datasets adopt this framing. While program-level analysis is also important,
> it addresses a different challenge. COMPOTE is designed to fit this established setting
> while addressing its primary bottleneck: the lack of real-world, compilable binary data. The
> system preserves the original function exactly by overwriting the body after every LLM step,
> ensuring no semantic drift is introduced by the scaffolding. Our contribution is therefore not
> a redefinition of the task, but a real-world alternative to existing synthetic datasets at the
> same granularity. We believe there remains significant room for progress in function-level
> binary vulnerability detection, and both the COMPOTE tool and the CompRealVul_Bin
> dataset provide resources that meaningfully support and advance research within this widely
> adopted problem formulation.
>
> **Response for weakness 2**
>
> Thank you for the suggestion. We agree that showing a concrete example is helpful for
> illustrating how COMPOTE preserves the original code. To make this explicit, we added
> to the appendix a clear before-and-after example of a real vulnerable function prior to
> compilation (appendix Section B.1 figure 6). This example will show the untouched original
> function alongside the generated scaffolding, demonstrating that COMPOTE does not modify
> the function body and only adds the minimal wrapper needed for successful compilation.
>
> **Response for question 1**
>
> CompRealVul is constructed from ten established real-world vulnerability datasets such
> as Devign, ReVeal, BigVul, DiverseVul, PrimeVul, CveFixes, and others. These datasets
> are all mined from real software repositories and vulnerability-fixing commits on platforms
> like GitHub. COMPOTE applies its wrapping process to these raw functions, producing
> compilable source and binary versions while preserving the original labels. For completeness
> and clarity, Section 4 (“CompRealVul Datasets”) now includes the full list of all datasets
> name incorporated into CompRealVul.
>
> **Response for question 2**
>
> COMPOTE is designed to strictly preserve the original target function with no modification
> to the function itself at all. The system enforces this by overwriting the function body with
> the original version after each LLM generation step, ensuring that only the surrounding
> scaffolding changes. Our goal is to support binary-level ML research, where preserved
> function semantics and guaranteed compilability are the key requirements.

---

### Official Review · Reviewer_gJRB · 2025-11-01

**Soundness:** 2
**Presentation:** 2
**Contribution:** 2
**Rating:** 2
**Confidence:** 3

**Summary:**

This paper aims to propose a LLM-based automated pipeline that takes incomplete or non-compilable real-world C code snippets, automatically wraps, fixes, and compiles them into programs, outputting two datasets:one is compilable C source files, the other is their corresponding compiled binaries, total yielding over 18K compilable programs (approximately 37% success rate from 49K raw functions).

Claimed Contributions:

- They developed a system (COMPOTE) for LLM-guided code wrapping and error correction using iterative compiling feedback.

- They claimed that this is the first large-scale, compilable, real-world binary dataset for vulnerability detection.

- They claimed to empirically have show n that models trained only on synthetic datasets perform poorly on real-world binaries; models trained or fine-tuned on CompRealVul generalize better to real code.

- They claimed that combining both real and synthetic data provides the best generalization.

**Strengths:**

They tackled a real data bottleneck in binary-level vulnerability ML.

They work enables binary-level ML training for models like RNN and ModernBERT - a previously missing benchmark.

**Weaknesses:**

1. Limited novelty: using GPT/LLMs to fix code for compilation has been explored before. The novelty mostly lies in pipeline integration.

2. Evaluation design: No direct CompRealVul vs. Juliet-only comparison - results emphasize "Combined" training, not a pure head-to-head. Sample imbalance makes cross-dataset metrics hard to interpret. Does not evaluate whether COMPOTE introduces semantic drift during wrapping.

**Questions:**

1. The paper leaves unanswered whether CompRealVul truly outperforms other "real-world" datasets.

2. The paper doesn't test how much the performance improvement comes from realism vs. data curation.

---

> ### Author Response · Authors · 2025-11-21
>
> **Response For weakness 1:**
>
> Our goal is not to introduce a new LLM technique, but to solve the missing real world data bottleneck in binary-level vulnerability research. COMPOTE is an effort to address that problem, while ensuring that (i) the exact original function is preserved, (ii)  iterative compiler feedback is integrated, and (iii) it scales to tens of thousands of real-world vulnerability snippets to produce a large, compilable, binary benchmark. This is what enabled a dataset such as CompRealVul\_Bin - the equivalent of which did not previously exist in this domain - to be created for the first time.
>
> **Response For weakness 2:**
>
> Thank you for raising this points.
>
> - **Direct CompRealVul vs. Juliet-only comparison.** The paper includes the head-to-head comparisons:
> Train Juliet → Test Juliet,
> Train Juliet → Test CompRealVul,
> Train CompRealVul → Test CompRealVul.
> These results appear in Table 3 and Table 4. They show that Juliet-only training performs well only on Juliet, but drops sharply on CompRealVul (for example, ModernBERT falls from 98.97 F1 to 35.47 F1). This pure head-to-head comparison demonstrates that Juliet’s synthetic patterns do not reflect real-world binary complexity.
> At the same time, the CompRealVul-only results show that real-world vulnerabilities are inherently harder to learn, which makes a real-world dataset essential for training models that capture realistic patterns. This highlights the need for CompRealVul not only as an evaluation set but as a training resource that exposes models to authentic structures absent from Juliet.
> The combined training setup is included because Juliet and CompRealVul each contribute complementary strengths: Juliet provides controlled patterns, while CompRealVul brings real-world variability needed for robust generalization.
> We have added clarifications to section 5.2.1 to make this point clearer:
> "We evaluated our models under four distinct train-test regimes that cover both real-synthetic comparisons and a combined setting. Specifically, we trained on CompRealVul and tested on CompRealVul to measure performance on real-world binaries; trained on Juliet and tested on Juliet to capture performance on synthetic data; trained on a Juliet subset and tested on CompRealVul to assess synthetic-real generalization; and finally trained on the combined dataset and tested on CompRealVul to examine mixing synthetic and real data. Across these four experiments, each architecture was trained or fine-tuned, resulting in 32 total model runs."
> - **semantic drift
> during wrapping.** COMPOTE does not modify the original function. The wrapper is injected only around the function to create a standalone compilable program. The system preserves the function’s exact body by overwriting it after every LLM-generated step, ensuring that no semantic drift is introduced by the scaffolding. Thus, the function’s semantics remain unchanged throughout the process.
>
> **Response For question 1:**
>
> Thank you for raising this point. The reason we wrote this paper is exactly because there are no other "real-world" binary level datasets. All existing real-world datasets are released only as source code and cannot be compiled, making direct comparison impossible. This absence is exactly the gap that CompRealVul addresses by providing a realistic dataset that can be compiled for binary-level evaluation.
> For completeness, we compare against Juliet, the well-known synthetic dataset that is compilable. Our results show that models trained solely on Juliet fail to generalize to real-world binaries, underscoring the need for a real-world, compilable dataset like CompRealVul.
>
> **Response For question 2:**
>
> We believe this concern stems from an assumption that COMPOTE’s wrapping or fixing may have altered the underlying semantics of the functions. In our case, the evaluation is performed only on the compiled functions whose internal logic has not been modified by the pipeline. The LLM-driven corrections are limited to making the code compilable (e.g., adding headers, completing missing return statements, resolving trivial syntax issues), but we do not alter the core function bodies or inject new logic.
> Because the model is trained and evaluated on these post-compilation binaries-which preserve the original function body-we argue that the performance improvements cannot be attributed to “data curation” in the sense of semantic simplification or rewriting. Instead, the gains come from the dataset capturing the complexity and distribution of real-world C code, which synthetic datasets like Juliet do not reflect.
> That said, we agree that separating the effects of realism from general code normalization could be interesting. But in our current workflow, the curated aspects do not change the semantics of the functions being learned. Therefore, the improved generalization originates primarily from the real-world nature of the dataset, not from simplification introduced by COMPOTE.

---

### Note · Program_Chairs · 2026-01-17
**Submission Desk Rejected by Program Chairs**

The following references in this submission do not refer to real documents and/or have major errors in bibliographic information:

 Giovanni Grieco, Lorenzo Grinblat, Marino Miculan, et al. Toward massive automated binary analysis. In Proceedings of the ACM SIGSAC Conference on Computer and Communications Security (CCS), 2016.